# First Report of Necrophilia in the Form of Necrocoitus among Insects, Involving Two Male *Cryptotympana atrata* (Fabricius, 1775) Cicadas

**DOI:** 10.3390/insects12010020

**Published:** 2020-12-31

**Authors:** Ji-Shen Wang, Victor Benno Meyer-Rochow

**Affiliations:** 1College of Agriculture and Biological Sciences, Dali University, Dali 671003, China; wangjishen826@sina.com; 2Department of Ecology and Genetics, Oulu University, SF-90140 Oulu, Finland; 3Agricultural Science and Technology Research Institute, Andong National University, Andong GB36729, Korea

**Keywords:** auchenorrhyncha, homosexual, mating, aberrant behaviour, thanatophilia

## Abstract

**Simple Summary:**

It has been reported that members from all vertebrate classes occasionally attempt copulations with dead male or female individuals, but until now no reliable report exists that insects too may show such necrophilic behaviour. We observed and described how a male cicada from Jinxiu town in southern China repeatedly tried to mate with a dead male conspecific. There are several ways to explain this unusual behaviour, but we believe that possibly a lack of females, the somewhat larger than normal size of the dead male individual and the latter’s passivity have been involved. This is the first report of a male insect mounting a dead male and attempting to copulate with it.

**Abstract:**

The unusual case of a male *Cryptotympana atrata* cicada from China attempting to mate with a dead male conspecific is described and illustrated. Although hitherto unreported, necrophilic behaviour in the form of an attempted necrocoitus, involving dead male or female corpses, may not be as isolated a case as has been previously assumed, but it does not seem to have been mentioned earlier in the entomological literature. Although the described behaviour could have been an expression of a lack of opportunity to locate a cooperative female, several other possibilities, referred to in the Discussion, exist and should not be dismissed.

## 1. Introduction

The fact that males of vertebrate animals sometimes copulate with dead females has been reported from all vertebrate classes. One of the authors (V.B.M.-R.) has seen male guppy fish (*Lebistes reticulatus*) courting a dead and bloated female and making repeated and vigorous attempts to insert the gonopodium into the female’s genital opening. On several occasions he also came across toads (*Bufo bufo*) in amplexus, in which the male was clasping the dead body of a female. For reptiles, mating with dead females has been described in lizards [1] and snakes [2], and for birds there exist published reports involving pigeons [3], sand martins [4] and penguins [5]. Male mammals are also known to mate occasionally with dead individuals. In humans such a behaviour is termed necrophilia and punishable in most countries as an unacceptable sexual deviation. Although copulations between a male insect and a dead male or female may have been observed before, there does not seem to be any published record of such an unusual behaviour, which is why we feel our observation is worth being made public.

For insects the term “necrophilia” is somewhat misleading, because appreciating dead bodies as a source of food or a place in connection with brood rearing could be covered under the term and is something not at all rare among insects. On the other hand, the more specific term “necrocoitus” removes the aforementioned possibility of a misunderstanding and could be applied to what we are describing here.

## 2. Materials and Methods

The insects were encountered during field research on a sunny afternoon on 22 July 2015, in Jinxiu Town (Guangxi, China) and observed for an unrecorded period of time. Photographs, shown in Figure 1a,b, were taken by Ji-Shen Wang with a Nikon D7000 digital camera in conjunction with an AF-S VR micro-Nikkor 105 mm f/2.8G IF-ED lens. Photographs of male and female *Cryptotympana atrata* cicadas, shown in Figure 1c, were taken by Zhi-Liang Wang with a Canon 760D digital camera (Tokyo, Japan) in conjunction with a Canon EF 100 mm f/2.8L Macro IS USM lens (Tokyo, Japan).

## 3. Results

During field research on a sunny afternoon on 22 July 2015, in Jinxiu Town (Guangxi, China), we noticed a seemingly normal and uninjured male cicada of the species *Cryptotympana atrata* (Fabricius, 1775) copulating with a dead conspecific that was identified as a another but slightly larger male than the live male (Figure 1a,b). The lifelessness of the dead individual was confirmed by its unresponsiveness to prodding, its stiff legs and its inability to move. Unfortunately, we did not witness over which distance the male detected the dead individual and how the former recognized the dead individual as a potential sex partner. In most insects, females (whether attracted to the vicinity of male conspecifics or not) convey to a male their willingness to mate and by using chemical or visual signals as key-stimuli encourage the males to mount them. In cicadas, it is the sound of the males that leads the females to near where the males are singing, but then the males must locate the female and initiate copulation.

In the case we observed, penetration was attempted while the dead male was mounted. Successful penetration was not seen, which is perhaps not surprising given the dead individual’s lack of cooperation and stiff body. After repeated attempts to penetrate, the active male abandoned the dead sex partner and flew off. Neither individual was collected and only the photographs served to identify the species.

## 4. Discussion

In the case we observed, we assume that the live male, using the sound emitted by its well-developed tymbal organs to attract females to the tree it was sitting in, recognized the dead conspecific on the branch by vision either as the former was approaching the tree and then died or, more likely, as it was already quietly resting on the branch, being dead. This suggests that the eyes of the male must have had quite a remarkable acuity and been able to spot the other individual over a certain distance. For humans to distinguish the sexes of cicadas is not too big a problem on account of the female’s more pointed abdominal shape, but for a male to spot that difference would have been difficult (Figure 1c). Cicadas and spittle bugs are members of the hemipteran suborder Auchenorrhyncha, and studies of the eyes of the spittle bug *Philaenus spumarius* (L.) by Keskinen and Meyer-Rochow [7] and the cicada *Psaltoda moerens* by Ribi & Zeil [8] have demonstrated that these insects possess eyes with a retinal organization that suggests visual acuities not too different from those of flower-visiting insects like, for example, bumblebees [9]. A visual detection of a conspecific even over a distance of at least one metre should therefore have been entirely possible, unless of course the live male walked upon it by chance.

Whether cicada females emit some pheromone or differ in terms of odour from the males has apparently never been investigated, presumably because researchers deemed that to be very unlikely. In fact, we do not know how cicadas identify each other by gender and if the male that we observed might not equally well have approached a conspecific female, dead or alive, had there been one around. A drake, for example, has been reported to engage in homosexual behaviour with a male corpse [10]. The report of a penguin engaging in necrocoitus [5], assumed to have involved a dead female, might also have been with a dead male as the sexes in penguins are morphologically almost indistinguishable from each other. A remote, but perhaps unlikely possibility is that the male cicada was alive when the other male started to copulate with it, but that later, while still in the process of being coupled, it died for reasons unknown. That the two individuals became inseparably hooked to one another can be ruled out as the live male could voluntarily disengage from the dead individual after carefully having been taken off the branch with its partner and placed on a stone for a more detailed photograph of the two apposed males’ genitalia (Figure 1b). 

What in our specific case made the male cicada copulate with another and moreover dead male is difficult to answer as we do not know how long the latter had been dead and what criteria cicadas use to recognize death. We can certainly rule out that this was a “practice run”, suggested for some cases of homosexual activity in mammals [11]. That aggression and a subsequent attack could have turned into a mating episode is possible but unlikely as combat between male cicadas is rare. With the dead individual’s slightly larger body resembling that of a female (and the lack of a defensive reaction when approached by the live male), a supernormal visual releasing stimulus as described by Kral [12] in the *European Journal of Entomology* for exceptionally attractive sex partners, could possibly have been involved. Not to be dismissed are also the possibilities that the male could have been confused and thought it was mounting a live female, or that there are cicada males that are attracted to other males and would be rebuffed by living (heterosexual) males. However, a lack of females (although we have no data to confirm this) is deemed by us to have led to the erroneous mating with a passive, non-resistant and somewhat larger male conspecific individual. Assuming a more suitable sex partner was not available, this seems to us to have been the most likely explanation for the scenario we observed and described in this research note.

## 5. Conclusions

Although rare, necrophilic behaviour in the form of necrocoitus has been reported from all vertebrate classes. Our report shows for the first time that that there are also insects (such as the cicadas we observed), which can engage in this kind of unusual behaviour. However, no single explanation for this behaviour can currently be given as there are various possibilities that could have been involved.

## Figures and Tables

**Figure 1 insects-12-00020-f001:**
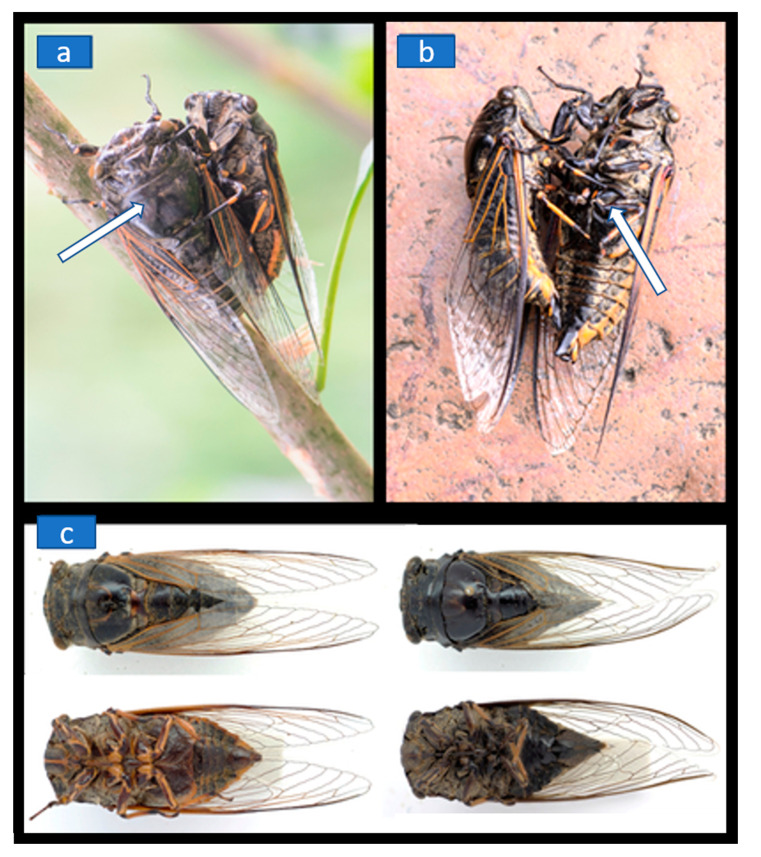
Two male *Cryptotympana atrata* cicadas engaged in mating on a branch (**a**) and then placed on a stone for closer inspection especially of the male genitalia (**b**). The white arrow points to the dead male. (**c**) The upper row shows the dorsal side of a male (left) and female (right), while the bottom row shows the corresponding ventral sides. According to Hayashi [6] the species is very variable and geographic and/or individual variations within *C. atrata* exist. Body lengths for both genders have been reported to be 36–46 mm (females are often slightly larger than males).

## Data Availability

No additional data available.

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
