# Peer review of "First Report of Necrophilia in the Form of Necrocoitus among Insects, Involving Two Male Cryptotympana atrata (Fabricius, 1775) Cicadas"

_insects, 2020, doi:10.3390/insects12010020_

Round 1

Reviewer 1 Report

REVIEW of the manuscript Insects-1059064 entitled

First report of necrophilia in the form of necrocoitus among insects, involving two male Cryptotympana atrata (Fabricius, 1775) cicadas

by Ji-Shen Wang and Victor Benno Meyer-Rochow

GENERAL COMMENTS

This is a short, but very interesting manuscript presenting an exciting, novel observation that shows invertebrates can behave with regard to necrophilia just like the vertebrates for which this aberrant behaviour has been described before. The Authors report for the first time that this form of sexual aberration is not restricted to vertebrates, but can also occur in insects.

As the only minor criticism, I can mention that it would have been nice to know, if penetration was achieved during the observed necrocoitus. But it is only a very minor point.

SPECIFIC COMMENTS

row 27:

'[3]Slavid & Taylor, 1987)'

should be replaced by

'[3](Slavid & Taylor, 1987)'

---

rows 42-43:

'identified as a another but'

should be replaced by

'identified as another but'

---

rows 53-54:

In the caption for Figures 1-3 the Authors wrote:

'The white arrow points to the dead male.'

However, this white arrow is lacking in Figures 1 and 2. It should be added to both figures.

---

In row 55 'to Hayashi [11]' should be replaced by 'to Hayshi [11]'

OR

In row 133 '11. Hayshi, M. A' should be replaced by '11. Hayashi, M. A'

---

In rows 59-62 the Authors wrote:

'... the male, using the sound emitted by its well-developed tymbal organs to attract the female to the tree it was sitting in, recognized the conspecific by vision either as the latter was approaching the tree or as it was already quietly resting on a branch of the tree.'

I do not understand, how the dead male conspecific could approach the tree? Perhaps, the cited sentence part should be rewritten as follows:

'... the male, using the sound emitted by its well-developed tymbal organs to attract the female to the tree it was sitting in, recognized the DEAD conspecific ON THE TREE BRANCH by vision either as the FORMER was approaching the tree or as it was already quietly resting on THE branch.'

Author Response

We thank this and the other 3 reviewers for having made valuable suggestions and recommended 'minor revision'.

We followed this reviewer's in all points and wish to thank him/her for having pointed out the misleading sentence on rows 59-62. We have adopted the sentence suggested by the reviewer. 

Reviewer 2 Report

This paper describes a remarkable observation, and one that will surely be noticed by readers outside the field. I am satisfied that the authors saw what they claim to see, and feel it absolutely merits publication. However, care must be taken when describing the observation, as extraordinary claims require extraordinary evidence. Also, some grammar/style edits are needed.

Major concerns:
16 and 92-96 - I personally feel the "sexual frustration" hypothesis is too much of an anthropomorphization. Is it possible the cicada was just confused and thought it was mounting a live female? Confusion, frustration, or a genetic defect in sensory systems or mating behavior would be equally likely. Or could it have been genuinely homosexual? Perhaps some cicada males are exclusively attracted to other males? Perhaps such a male would be rebuffed by living [heterosexual] males, and thus only able to mount a corpse? Importantly, the authors do not provide evidence for frustration, such as a noted lack of females in the area, or evidence that the male was a virgin. In absence of evidence, I suggest the authors do not choose an explanation, but lay out the options and perhaps a course for future researchers to confirm which is more likely.

61-62 This implies "the latter" male, which I assume means the cicada being penetrated, was alive before the mating event. You have no evidence of this.

The discussion does not have a clear flow. 72-85 for example addresses too many unrelated subjects. Please structure the discussion more carefully. Paragraph 1: The penetrating male finding a conspecific at a distance. P2: Could this male have thought the other male was a female, and can cicadas identify each other by gender? P3: Could the penetrating male have known the other male was dead, or was the other male alive at the time? P4: Was sexual penetration involved, and could the penetrating male have stopped the coitus at any time? P5: What could have motivated this male to engage in homosexual necrocoitus? P6: What future observations could confirm or reject the hypothesized motivations?

I do not see white arrows in Figures 1 or 2. Please fix them.

In figure 2, the genitalia are not clearly labelled. It is not clear or described in the text whether or not the penetrating male's sex organ was everted at any time. This is critical: was the male mounting, or penetrating?

Lastly, I am curious as to what happened to the two males after this observation. Were they both collected? Are they stored in a museum or private collection? Did the living male behave normally? Did it fly away?

Major grammar/Style edit
The references should either be [number] or (author, year), not both, unless this journal has a very unique style, or the authors are waiting for paper acceptance before formatting the citations.

Minor Grammar edits:
16 add a comma after "corpses"
22 change "one of us" to "one of the authors"
22 "seen a male"
25 add comma after "reptiles"
26 add comma after citation 2
29 replace "individuals and with regard to humans" with "individuals. In humans,"
40 add comma after "2015"
47-49 I am a bit confused here. Are females attracted to the vicinity of males, or the males themselves? Do they use chemical or visual signals only to encourage males to approach them, or also to convey willingness to mate? Please rewrite this as two separate sentences, or one sentence with at least one comma to mark a pause. Or, replace this and the following sentence with a very simple description of what happens, in as many separate sentences as necessary to maximize clarity.
50 replace "but it is then for individual males to locate the female and to initiate" with "but then the males must locate the female and initiate"
56 delete the comma after "C. atrata"
59 "the male" is ambiguous, as there are two males.
60 replace "the female" with "females"
62 delete "however"
63-66 the sentence "For humans… difficult (Fig. 3)" should go at the end of the paragraph or in a later paragraph, perhaps the one where you mention cicada pheromones. First you must address whether or not cicadas can spot conspecifics at the distance of 1 meter, then discuss identifying gender at close range. I would also add that speculating on what happened before mating is just speculation: you have no evidence, so cannot state at what distance the living male first noticed the dead male. Perhaps he walked upon it, rather than spotting it from a meter away.
66 add comma after "Auchenorrhyncha"
68 replace "as well as" with "and"
70 delete ", for example," and consider adding text explaining that flower-visiting insects have good vision
70 Start last sentence with "Visual," and maybe
73 replace "as that to researchers must have been deemed" with "because researchers deemed that"
75-76 Replace "Regarding drakes, for example, it had been reported that they engaged" with "Drakes have been reported to engage"   But, if this reference is only about a single drake, then say "Drake" and "male corpse." If this behavior has been observed repeatedly, then say "drakes" and "male corpses."
77 The drake and penguin cases should be in separate sentences
81 delete "a reason or"
86 replace "another (but dead)" with "another, dead"
87 put "practice run" in quotations and cite the source that suggested this for mammals
89 "combat between male cicadas is rare" [combat is an uncountable noun]
90-91 This could also be at least two sentences. First, state clearly that the dead male was larger than the typical female, and that females are typically larger than males in this species, and so by size alone it could have appeared as a female. Then, state that, as it was dead, it would not have been able to produce a defensive reaction the way females do when uninterested in mating. Finally define "supernormal visual releasing stimulus," as not all readers will want to read Kral's paper to learn what it means.

Author Response

We thank this reviewer and the other three for their helpful and valuable comments and for recommending 'Minor Revision'.

This reviewer expressed his concern about our attempt to explain the behaviour and wrote  I personally feel the "sexual frustration" hypothesis is too much of an anthropomorphization."

We agree and have changed that in the Abstract and in the paper itself

"Is it possible the cicada was just confused and thought it was mounting a live female? Confusion, frustration, or a genetic defect in sensory systems or mating behavior would be equally likely. Or could it have been genuinely homosexual? Perhaps some cicada males are exclusively attracted to other males? Perhaps such a male would be rebuffed by living [heterosexual] males, and thus only able to mount a corpse? Importantly, the authors do not provide evidence for frustration, such as a noted lack of females in the area...."

Reviewer is correct and we added other possibilities in the Discussion.

Like the the reviewers this one noticed that the white arrows were missing in the figure. They have now been inserted.

S/he wanted us to add information what happened to the two cicadas and if they were collected. We have added that information. We also explained that 'penetration' was not observed (and why it would have been difficult), but that it was attempted while the active male was mounting the dead male.

We do not feel it is necessary to indicate the genitalia in fires 1 and 2 as entomologists would all be familiar with male insect anatomy.

References to the works of others were corrected and now follow the style of the journal as also seen in other published papers in that journal.

This is how we responded to what the reviewer termed minor Grammar edits:

16 add a comma after "corpses"    DONE
22 change "one of us" to "one of the authors"    DONE
22 "seen a male"    DONE
25 add comma after "reptiles"    DONE
26 add comma after citation 2    DONE
29 replace "individuals and with regard to humans" with "individuals. In humans,"
40 add comma after "2015"    DONE
47-49 I am a bit confused here. Are females attracted to the vicinity of males, or the males themselves? Do they use chemical or visual signals only to encourage males to approach them, or also to convey willingness to mate? Please rewrite this as two separate sentences, or one sentence with at least one comma to mark a pause. Or, replace this and the following sentence with a very simple description of what happens, in as many separate sentences as necessary to maximize clarity.
50 replace "but it is then for individual males to locate the female and to initiate" with "but then the males must locate the female and initiate"    THIS HAS BEEN REWRITTEN   
59 "the male" is ambiguous, as there are two males.    DONE
60 replace "the female" with "females"    DONE
62 delete "however"    DONE
63-66 the sentence "For humans… difficult (Fig. 3)" should go at the end of the paragraph or in a later paragraph, perhaps the one where you mention cicada pheromones. First you must address whether or not cicadas can spot conspecifics at the distance of 1 meter, then discuss identifying gender at close range. I would also add that speculating on what happened before mating is just speculation: you have no evidence, so cannot state at what distance the living male first noticed the dead male. Perhaps he walked upon it, rather than spotting it from a meter away.  THIS HAS BEEN CHANGED AND THE LAST SENTENCE THAT THE REVEWER SUGGESTED HAS BEEN ADOPTED.
66 add comma after "Auchenorrhyncha"    DONE
68 replace "as well as" with "and"    DONE
70 delete ", for example," and consider adding text explaining that flower-visiting insects have good vision  WE FELT THIS WAS NOT NECESSARY
70 Start last sentence with "Visual," and maybe
73 replace "as that to researchers must have been deemed" with "because researchers deemed that"    DONE
75-76 Replace "Regarding drakes, for example, it had been reported that they engaged" with "Drakes have been reported to engage"   But, if this reference is only about a single drake, then say "Drake" and "male corpse." If this behavior has been observed repeatedly, then say "drakes" and "male corpses."    YES, DONE
77 The drake and penguin cases should be in separate sentences    DONE
81 delete "a reason or"    DONE
86 replace "another (but dead)" with "another, dead"    DONE
87 put "practice run" in quotations and cite the source that suggested this for mammals   DONE AND REFERENCE HAS BEEN ADDED
89 "combat between male cicadas is rare" [combat is an uncountable noun] THANKS

This reviewer was the only one of the 4 reviwers who felt that the discussion did not have a clear flow and suggested some changes. We did make some changes, but we like the flow. The Discussion is not long and we kindly ask the reviewer to accept that authors have different preferences. e certainly toned down our interpretation of the behaviour, added alternatives and inserted some explanatory remark regarding Kral's 'super normal stimulus'.

Reviewer 3 Report

Changes have addressed concerns.  Two minor English adjustments needed.

Line 28 ...to occasionally mate... should be .. to mate occasionally... so the infinitive is not split.

Line 54  In Fig. 3, the upper row... add the comma after Fig. 3.

Author Response

We thank the reviewer for his/her supportive comments, which were taken into consideration during the  'minor revision'.

Reviewer 4 Report

The paper by Wang and Meyer-Rochow in a short report that could be of some interest based on the fact that it is the first report of necrocoitus in insects. I have only some minor suggestions:

1-The term "necrophilia" identifies a sexual deviation / perversion with psychological implications. I don't think it's applicable to insects and  I agree with the authors that "necrocoitus" is a better term.  Therefore,  I recommend deleting "necrophilia" from the title.

2-For similar reasons I recommend not to use "frustration" (line 16, Abstract; line 93, Discussio)

3-Results, line 40. Delete During …. sunny afternoon. It is irrelevant.

4-Discussion. Lines 59-71. This part is relevant but it could be easuly reduced.

Line 92. Delete "in this journal.

5-Figures. Sorry, I don't see any "white arrow"

Author Response

We thank this and the other three reviewers for their supportive comments and suggestion of 'Minor Revision'. Here are our responses:

1-The term "necrophilia" identifies a sexual deviation / perversion with psychological implications. I don't think it's applicable to insects and  I agree with the authors that "necrocoitus" is a better term.  Therefore,  I recommend deleting "necrophilia" from the title.

With all due respect, we disagree. In Entomology (and in fact in connection with other animals as well) the term 'necrophilia' is accepted quite unlike the situationn for human behaviour where (and the reviewer is correct) the term describesn a "perversion with psychologocal implications". For insects it is more complicated, because 'necrophilia' has also been used for species that seek out dead bodies to deposit their eggs on or to devour.

2-For similar reasons I recommend not to use "frustration" (line 16, Abstract; line 93, Discussio)

Here we agree and we have changed sentences accordingly.

3-Results, line 40. Delete During …. sunny afternoon. It is irrelevant.  We disagree, because cicadas are most active during sunny afternoons.

4-Discussion. Lines 59-71. This part is relevant but it could be easuly reduced. DONE

Line 92. Delete "in this journal.  DONDE

5-Figures. Sorry, I don't see any "white arrow" Reviewer is correct: Arrows have ow been added.

This manuscript is a resubmission of an earlier submission. The following is a list of the peer review reports and author responses from that submission.

Round 1

Reviewer 1 Report

Delete lines 21 to 30. Use references related to insects.

Delete lines 61 to 64. Use references related to insects.

Delete lines 64 to 70. This discussion is not relevant, it should be excluded.

Delete lines 74 to 78. Use references related to insects.

Line 78 to 81. The observations are inconsistent for publication, as they are based on hypotheses and not on actual data observed or studied.

Line 88 to 93. This information needs data obtained in the field to use, and no field data was obtained.

As this is a single observation, without prior information from the field, without observation data of the species' behavior, I do not recommend publication.

The article must be disapproved.

Reviewer 2 Report

An odd behavior documented with images.  Just a few minor suggestions.

Line 13 you may want to add authority (Fabricius, 1775) for species for those finding the abstract.

Line 14 change to ...a dead conspecific male is described... Add "male" as this is a specific part of the discussion.

Line 27 you may want to add species names of taxa involved as you had listed species names earlier (e.g. lines 22 and 24).  Same for line 74.

Line 28 change to ...to mate occasionally... you split the infinitive.

Line 43 change to ...witness over what distance...

Line 58 "timbal" is the correct spelling of the sound organ in cicadas not "tymbal", change to ...to the tree in which it was sitting,...

Line 69 change to ...Visual detection of...

The final paragraph on size determination by the cicada is not supported by the data.  Hayashi (1982 Bull. Kiyakyushu Mus. Nat. Hist. 7:1-109, available online) listed a range of 36-46 mm for the species in his revision of the genus.  Both sexes are reported here with smaller body lengths but we are increasing the range of body sizes for the species.  That makes it unlikely that a male is searching for a slightly smaller individual with which to mate.  I also have examples in my collection where females are larger than males.